# Evaluation of Skin Wound Healing with Biosheets Containing Somatic Stem Cells in a Dog Model: A Pilot Study

**DOI:** 10.3390/bioengineering11050435

**Published:** 2024-04-28

**Authors:** Noritaka Maeta, Ryosuke Iwai, Hiroshi Takemitsu, Natsuki Akashi, Masahiro Miyabe, Marina Funayama-Iwai, Yasuhide Nakayama

**Affiliations:** 1Faculty of Veterinary Medicine, Okayama University of Science, 1-3 Ikoi-no-Oka, Imabari 794-8555, Ehime, Japan; 2Institute of Frontier Science and Technology, Okayama University of Science, 1-1 Ridaicho, Kita-ku 700-0005, Okayama, Japan; 3Department of Comparative Animal Science, College of Life Science, Kurashiki University of Science and the Arts, 2640 Nishinoura, Tsurajima, Kurashiki 712-8505, Okayama, Japan; 4Japan Society for the Promotion of Science, 1-1 Ridaicho, Kita-ku 700-0005, Okayama, Japan; 5Osaka Laboratory, Biotube Co., Ltd., 3-10-1 Senriyama-Higashi, Osaka 565-0842, Suita, Japan

**Keywords:** in-body tissue architecture(iBTA), biosheet, somatic stem cells, skin wound healing, tissue engineering

## Abstract

The administration of mesenchymal stem cells (MSCs) has a positive effect on wound healing; however, the lack of adequate MSC engraftment at the wound site is a major limiting factor in current MSC-based therapies. In this study, a biosheet prepared using in-body tissue architecture (iBTA) was used as a material to address these problems. This study aimed to assess and evaluate whether biosheets containing somatic stem cells would affect the wound healing process in dogs. Biosheets were prepared by subcutaneously embedding molds in beagles. These were then evaluated grossly and histologically, and the mRNA expression of inflammatory cytokines, interleukins, and Nanog was examined in some biosheets. Skin defects were created on the skin of the beagles to which the biosheets were applied. The wound healing processes of the biosheet and control (no biosheet application) groups were compared for 8 weeks. Nanog mRNA was expressed in the biosheets, and SSEA4/CD105 positive cells were observed histologically. Although the wound contraction rates differed significantly in the first week, the biosheet group tended to heal faster than the control group. This study revealed that biosheets containing somatic stem cells may have a positive effect on wound healing.

## 1. Introduction

Mesenchymal stem cells (MSCs) are multipotent cells that can self-renew and differentiate into multiple tissue-forming cell lineages. Osteoblasts, adipocytes, chondrocytes, tenocytes, neurons, hepatocytes, and myocytes are some examples of these lineages [1,2,3,4]. Mesenchymal stem cells are derived from several tissues, including bone marrow, adipose tissue, dermal skin tissue, Wharton’s jelly, and the placenta [1,2,3,4]. Furthermore, they are also characterized by the positive expression of surface markers, such as CD73, CD90, and CD105, and negative expression markers, such as CD79α, CD34, CD45, and HLA-DR [5]. Therapeutic strategies using MSCs can be broadly classified into cell replacement therapy, which utilizes the pluripotency of MSCs to replace damaged tissues, and cell therapy, which utilizes the immunomodulatory, anti-inflammatory, and angiogenic effects of biologically active molecules secreted from MSCs.

Regenerative medicine using MSCs have been investigated in various tissues and organs, including bone, cartilage, muscle, nerves, myocardium, liver, cornea, trachea, and skin [2]. The role of MSCs in cutaneous wound healing is an area of active research, and their role and therapeutic effects on refractory chronic wounds caused by diabetes, burn injuries, and peripheral vascular diseases have been investigated [2,3,4]. Moreover, MSC administration has a positive effect on cutaneous wound healing by accelerating wound closure, enhancing re-epithelialization, increasing angiogenesis, promoting granulation tissue formation, modulating inflammation, and regulating extracellular matrix (ECM) remodeling [3]. In most previous studies, the delivery of MSCs has been performed by intradermal injection around the wound area. Although this method reportedly improves wound healing, poor engraftment efficiency and cell retention at the wound site have also been observed [6]. Furthermore, the lack of adequate MSC engraftment at the wound site is a major limiting factor in current MSCs-based therapies [7]. Therefore, alternative methods are required to efficiently deliver MSCs to the wound area and allow their survival.

The efficacy of stem cell-based therapies, such as MSCs for skin wound healing, has been reported; however, low rates of engraftment and cell retention at the wound site have been problematic [6,7]. In addition to the topical administration of MSCs, other methods, such as matrigel, fibrin polymers, and hydrogel seeding, have been reported to address these issues. Nevertheless, these methods may adversely affect cell viability [8,9,10,11]. Another approach involves the use of scaffold materials with MSCs, which may provide a suitable microenvironment for cell adhesion, proliferation, and differentiation [11]. Scaffolds can be made of natural biomaterials such as collagen and hyaluronic acid, which are major components of the extracellular matrix, and fibrin, a protein involved in coagulation that is highly biocompatible [4,12]. However, these methods require time-consuming and costly processes such as collection, separation, and, in some cases, MSC culture. Another disadvantage is that the scaffold material does not self-assemble.

Degradable biomedical elastomers (DBEs) are useful materials for tissue repair that promote skin wound healing [13]. Although DBEs are useful scaffolds for tissue repair, they do not contain somatic stem cells. Furthermore, the complexity of skin injuries necessitates that DBEs be combined with other functional materials for multifunctional repair.

An in vivo tissue engineering technology called in-body tissue architecture (iBTA) allows tissue preparation for autologous implantation without cell culture by simply implanting a mold under the patient’s skin [14,15]. Using the iBTA technique, tabular tissue structures such as biotubes [14,15] or heart valve-like tissues called biovalves [16,17] can be fabricated by means of simple subcutaneous embedding of molds. iBTA is based on encapsulation, which is a foreign body reaction, and collagenous tissues primarily occupy the interior of the embedded molds. Sato et al. histologically investigated the mechanism of biotube formation to clarify the formation process and found that the outer region of a biotube contains immature connective tissue consisting of type III collagen-containing primitive somatic stem cells expressing CD90 and SSEA4 [18]. Because of the tissue repair potential of these stem cells, biotubes may be useful alternatives to stem cell-containing materials in regenerative medicine.

Biosheets are developed using the iBTA technique, similar to biotubes. Additionally, during the formation process, biosheets may also integrate somatic stem cells within them during the fabrication process. The major advantage of biosheets is that all tissues are autologous and can be embedded subcutaneously to create scaffolds and somatic stem cell-containing tissues without any special processing. Owing to the inherent structural characteristics of a biosheet, the inclusion of somatic stem cells within its matrix may facilitate the efficient migration of these cells to the wound site when applied. Fully collagenized biosheets were found to be effective for the repair of abdominal and bladder wall defects [19,20]; however, the effects of immature biosheets comprising somatic stem cells on tissue healing were not assessed. Therefore, the purpose of this study was to assess and evaluate the wound healing capabilities of biosheets containing somatic stem cells in dogs.

## 2. Materials and Methods

### 2.1. Animals

This study strictly followed the animal care guidelines of the Laboratory Animal Center, Imabari Campus, Okayama University of Science, and was compliant with the Guide for the Care and Use of Laboratory Animals (8th ed.) (approval numbers: 2022-034, 11/5/2022; 2022-054, 11/7/2022).

Dogs were used in this study because they have sufficient subcutaneous space for implantation of the molds and sufficient skin space to perform skin wound healing comparisons.

Eight healthy adult beagles (1–3 years old, six males and two females, 7.9–10.2 kg) were enrolled in this study. The dogs were housed in the Laboratory Animal Center at the Imabari Campus, Okayama University of Science, under a 12:12 h light–dark cycle (light period: 8:00 a.m. to 8:00 p.m.). Temperature was maintained between 24 °C and 26 °C, and humidity was between 40 and 60%. The dogs were housed individually in stainless-steel cages with free access to water and food until approximately 12 h before anesthesia.

### 2.2. Preparation of Biosheets

The molds for biosheet preparation were embedded subcutaneously within the dogs. Three types of rectangular stainless-steel molds (width: 25.6 mm, height: 10.6 mm, length: 66 mm) with different pore shapes on the surface were used in this study (Figure 1). The molds with hexagonal pores on the surface were designated as O-shaped pore molds (Figure 1a,d), those with Y-shaped pores were designated as Y-shaped pore molds (Figure 1b,e), and those with a combination of O- and Y-shaped pores were designated as O/Y mixed-pore molds (Figure 1c,f). The experimental procedure was performed aseptically under general anesthesia. Needle catheters (22G) were placed in the right or left cephalic vein to administer medications and intravenous fluids. For the anesthetic protocol, antibiotics (cefmetazole: 20 mg/kg) and sedatives (medetomidine: 3 μg/kg) were administered intravenously, and analgesics (Robenacoxib: 2 mg/kg) were administered subcutaneously as premedication. Propofol (6 mg/kg) was administered intravenously, and the dogs were intubated using a tracheal tube. General anesthesia was maintained using isoflurane (2.0%; vaporizer setting) and oxygen (2 L/min). Ringer’s lactate solution was administered intravenously at 5 mL/kg/h during the procedure.

As shown in Figure 2, O- and Y-shaped pore biosheet molds were randomly embedded subcutaneously on the dorsal and ventral sides of the bilateral shoulders and abdominal regions of the four beagles (32 sites in total). The skin was incised, and the subcutaneous tissue was debrided using an electrocautery scalpel and forceps to create enough space for the molds. The molds were embedded weekly for 5 weeks, and after 6 weeks, the skin was incised using an electrocautery scalpel, and the subcutaneous tissue was peeled away to remove all molds. After the molds were removed from the skin, the biosheets were extracted and evaluated for appearance, weight, and strength and were assessed histologically.

### 2.3. Histological and Immunohistochemical Analysis

Biosheets were fixed in a 4% paraformaldehyde phosphate-buffered solution (pH 7.4) (Wako pure chemical; Osaka, Japan), embedded in paraffin, and sectioned at 3 to 5 µm thickness. The sections were subjected to routine hematoxylin and eosin (H&E) staining for nuclear detection and Masson’s trichrome staining for collagen fiber detection.

For immunohistochemical analysis, the deparaffinized sections were heated in Immunosaver solution (FUJIFILM Wako Pure Chemical Corporation, Osaka, Japan) at 90 °C for 1 h for antigen retrieval. Subsequently, the sections were washed in distilled water twice for 10 min and blocked in 1% bovine serum albumin (Wako pure chemicals) in PBS at 24–26 °C for 1 h and incubated with anti-SSEA4 mouse monoclonal antibody (1:100, ab16287; Abcam, Cambridge, UK) and anti-CD105 rabbit polyclonal antibody (1:200, bs-0579R; Bioss Inc., Boston, MA, USA) overnight at 4 °C. After being washed twice with distilled water for 10 min, the sections were incubated with Alexa Fluor 488 rabbit anti-mouse IgG antibody (1:1000, ab169345; Abcam) or Alexa Fluor 594 goat anti-rabbit IgG antibody (1:1000, ab150080; Abcam) at 24–26℃ for 2 h. DAPI (ProLong Gold Antifade Mountant with DAPI; Thermo Fisher Scientific, Inc., Waltham, MA, USA) was used as a nuclear counterstain. The sections were analyzed using fluorescence microscopy (ECLIPSE-Ti; Nikon Corporation, Tokyo, Japan).

### 2.4. Reverse Transcription and Quantitative Real-Time PCR (qRT-PCR) Analysis of mRNA

The biosheets were pulverized using a Power Masher II (KENIS, Osaka, Japan). Total RNA was extracted using the TRIzol reagent (Invitrogen, Waltham, MA, USA) according to the manufacturer’s protocol. Total RNA was quantified using a spectrophotometer. Total RNA (1 μg) was reverse transcribed at 37 °C for 15 min in 2 μL of ReverTra Ace (TOYOBO, Osaka, Japan) after inactivation of reverse transcription by heating at 98 °C for 3 min.

Quantitative RT-PCR was performed according to a previously reported protocol for determining mRNA expression in cultured cells [21,22]. The cDNA product was subjected to qPCR using a Real-Time PCR System 7500 (Applied Biosystems, Foster City, CA, USA) according to the manufacturer’s instructions. PCR was performed at 95 °C for 5 s and 60 °C for 34 s in 20 μL of buffer containing THUNDERBIRD^®^ SYBR™ qPCR Mix (TOYOBO) and 0.2 μM of each primer (Table 1). Each primer was designed based on GenBank information. The value of mRNA expression was calculated, expressed, and normalized to beta-actin. Each primer was used after confirmation that the PCR product contained the correct sequence. A linear amplification curve was established from serial dilutions of cDNA-containing plasmid DNA using quantitative measurements. Each reaction was performed in triplicate.

### 2.5. Generation and Treatment of Skin Defects

In four dogs, defects (50 mm × 50 mm) were created on the skin of both thoracic regions. The skin defect was cut through the entire skin layer using an electrocautery scalpel, the wound margin and subcutaneous tissue were debrided, and the cutaneous muscle and subcutaneous fat were removed to expose the fascia of the musculus latissimus dorsi (Figure 3a). The left side was used as the control, and a biosheet was applied to the right side. The right side was covered with the inside of the biosheet to cover the entire skin defect (Figure 3b). The biosheet and soft silicone wound dressing (Mepitel^®^ One: Mölnlycke Health Care, Tokyo, Japan) were sutured to the skin at the four corners of the wound edge and the center of the edge using nylon (Figure 3c). The wound was further covered with a wound dressing material (AQUACEL™ Ag Advantage: CONVATEC, Plus moist™ V; ZUIKO MEDICAL CORPORASTION, Osaka, Japan), and the wound dressing material and skin were secured with skin staples, covered with gauze, and fixed with a net bandage. The left side (control) was treated in the same manner, except that the biosheet was not applied to the skin defect. After the procedure, the wound was checked weekly for 8 weeks, and the wound dressing material was replaced.

### 2.6. Statistical Analysis

Statistical analyses were performed using GraphPad Prism 10 software (GraphPad Software, La Jolla, CA, USA). Nonparametric tests (Mann–Whitney U tests) were used to compare two groups, with *p* < 0.05 indicating a significant difference.

## 3. Results

### 3.1. Comparison of Biosheets Created Using Different Types of Molds

No adverse events, such as wound infection or deterioration of the general conditions of the dogs, were observed during the experimental period.

The biosheet of the 1-week group was mostly blood-like red, and the red portion decreased as the weeks progressed and turned into a peach color. The same trend was observed for both O- and Y-shaped pore types.

The lowest weight was observed in the 1-week group, whereas the 2-week group biosheet was the heaviest; the weight subsequently decreased in both the O- and Y-shaped pore types (Figure 4). A clear decrease in weight was noted in the Y-shaped pore types in the 3-week group. The O-shaped pores tended to be heavier than those of the Y-shaped pores.

Rectangular tubular tissue was cut longitudinally to make rectangular sheets, which were measured for strength. The 1-week group was jelly-like and soft and could not maintain a rectangular shape but stiffened in a time-dependent manner.

Histologically, the biosheet in the first week showed little collagen formation and was mostly comprised of fibrin tissue. Collagen formation began gradually in the second week, with the formation being faster in the Y-shaped pore-type biosheet. In contrast, the O-shaped pore biosheet was still mostly fibrin tissue, although collagen formation had begun. The biosheet created with Y-shaped pore types showed poor formation of both fibrin and collagen tissues in the 3-week group. From the third week onward, the proportion of collagen tissue increased and that of fibrin tissue decreased over time in both pore types; however, the Y-shaped pore-type biosheet lost fibrin tissue more rapidly than the O-shaped pore-type biosheet.

The results of macroscopic and histological evaluation suggested that biosheets prepared with O-shaped pore molds at 2 weeks post-implantation may be suitable for clinical use; therefore, the expression of inflammatory cytokines, interleukins, and Nanog mRNA was examined in the biosheets.

Total RNA was obtained from the biosheets prepared with the O-shaped pore mold at 2 weeks post-implantation. qPCR was used to examine the expression levels of cytokines in the biosheets (Figure 5). For the biosheet prepared with O-shaped pore molds, the expression of vascular endothelial growth factor (VEGF) was highest among the cytokines. Moderate levels of fibroblast growth factor (FGF) and platelet-derived growth factor (PDGF) expression were observed in O-shaped pores. Furthermore, low expression levels of hepatocyte growth factor (HGF) and brain-derived neurotrophic factor (BDNF) were observed in the O-shaped pores (Figure 5).

Interleukin mRNA expression in the biosheet prepared with the O-shaped pore-type mold increased in the order of IL-4, IL-10, and IL-6 (Figure 5).

The expression levels of canine Nanog mRNA in the biosheets prepared with O-shaped pore molds were examined by qRT-PCR (Figure 5). Nanog expression was clearly observed in biosheets prepared with O-shaped pore molds.

It was suggested that biosheets prepared with O-shaped pore molds at two weeks post-implantation might contain somatic stem cells. However, it was thought that cells might be more likely to invade the larger opening in the mold surface; therefore, a larger opening on the mold surface was created by combining O- and Y-shaped pore molds (O/Y mixed-pore type), and the properties of the biosheets produced two weeks post-implantation were compared macroscopically and histologically.

The O/Y mixed-pore biosheet molds were embedded in the dorsal and ventral sides of the unilateral shoulders and the dorsal side of the abdominal regions of the dog and removed 2 weeks after implantation.

Regardless of the mold pore shape, solid tissue formed in the mold two weeks after mold embedding (Figure 6a). The H&E staining revealed no significant accumulation of inflammatory cells in any tissue (Figure 6b). Masson’s trichrome staining showed that in the Y-shaped pore molds with relatively low porosity, collagen-rich tissue was formed, which was depicted by blue staining (Figure 6(c-1)). Conversely, the tissues formed in the O- or mixed O/Y-shaped pore molds with high porosity consisted of fibrin-like coagulum, which was stained light purple and dark red, and partially collagenous regions were present (Figure 6(c-2)). The collagen-rich tissue had a high cell density and was in the process of forming connective tissue with capillary angiogenesis (Figure 6(c1-1)); a few cells positive for both SSEA4 and CD105, mesenchymal stem cell markers, were observed in this tissue (Figure 6(c1-2)). By contrast, cells were observed in the fibrin-like coagulum (Figure 6(c2-1)), more than half of which were positive for both SSEA4 and CD105 (Figure 6(c2-2)). Here, a significantly larger number of SSEA4- and CD105-positive mesenchymal stem cells was observed in the O or O/Y mixed-pore molds than in the Y-shaped pore molds; therefore, the use of O/Y mixed-pore molds were selected for all subsequent transplantation experiments.

### 3.2. Comparison of the Effects of Biosheets on Wound Healing

Based on these results, the biosheet produced by the subcutaneous implantation of O/Y mixed-pore molds for 2 weeks was the most suitable for application in skin wound healing, and its effect on skin wound healing was investigated.

At 1 week post-implantation, the implanted biosheets were clearly visible (Figure 7). The biosheet-implanted group did not develop transient wound enlargement, as observed in the control group. In addition, there was no adhesion between the wound margin and the subcutaneous tissue in the control group, whereas there was adhesion in the biosheet-implanted group. Two weeks after implantation, the implanted biosheets were not clearly visible (Figure 7). In both groups, the wound edges were adherent to the subcutaneous tissue, and good granulation tissue formation was observed; however, wound contraction tended to be stronger in the biosheet-implanted group. Three weeks after implantation, the implanted biosheets could not be identified (Figure 7). Rapid wound contraction and clear epithelialization were observed in both groups. There was no obvious difference in the wound healing process between the two groups, but epithelialization tended to be delayed in the control group compared to the biosheet group; at 8 weeks after implantation, epithelialization was incomplete in two of four dogs in the control group. Adverse events, such as wound infection or deterioration of general condition, were not observed during the experimental period.

To objectively evaluate the differences between the two groups, the areas within the wound margins and granulation tissue areas were measured using Image J (Version 1.53t Image Processing and Analysis in Java) based on photographs of the wounds each week (Figure 8a), and the rates of wound contraction and epithelialization ratios were calculated and compared each week. The area within the wound margin was defined as the area inside and around the suture line that sutured the biosheet or soft silicone wound dressing to the skin. Additionally, no hair growth was observed (Figure 8a). Each area was measured five times, and the average values were compared (Table 2). The rate of wound contraction was significantly different (*p* < 0.05) in the first week and was not significantly different thereafter (*p* > 0.05); however, the biosheet group tended to contract faster than the control group up to the fifth week until the wound contraction was completed (Figure 8b). The epithelialization ratios were not significantly (*p* > 0.05) different at any time point; however, the biosheet group tended to epithelialize faster than the control group (Figure 8c).

## 4. Discussion

This study clarified that somatic stem cells are contained within the biosheets made from the molds used in this study. Furthermore, a larger opening on the surface of the mold may contain more somatic stem cells. Comparative wound healing studies using the prepared biosheets suggested that they may have a positive effect on wound healing. In addition, no side effects or complications were observed during the creation or use of the biosheets, suggesting that biosheets may be safe for in vivo use.

This is the first known study to use rectangular molds, as many previous studies made use of cylindrical molds [14,15,16,17,18,19]. This decision was made to allow for the creation of sheets with sufficient areas to cover the skin defects, as the diameter of a conventional cylindrical mold must be increased, thereby making the mold thicker. The thicker the mold, the more difficult it is to embed it in a limited space under the skin. Even if it can be embedded, the unevenness of the embedded area may cause complications, such as impaired local skin circulation and separation of skin sutures. The rectangular molds formed biosheets without problems and contained stem cells, indicating that the rectangular molds were useful for creating large-area sheets.

Three types of molds with different surface apertures were evaluated. The larger the mold surface aperture, the more somatic stem cells were included and the slower the replacement with collagen tissue. The process of biotube formation has been verified and is suggested to be as follows [12]: After the mold is embedded, the gap is filled with fluid from the subcutaneous tissue, fibrin is formed, and subcutaneous fibroblasts migrate in the mold. Growth factors derived from the fluid induce the formation of neoplastic tissue, and various cells, including somatic stem cells, are mobilized through blood flow from the blood vessels in the neoplastic tissue. Finally, the stem cells and fibroblasts differentiate into myoblasts, facilitating the formation of thick-walled tissues. The larger aperture area on the mold surface may allow more fluid to flow into the mold over a shorter period, facilitating the formation of fibrin and neoplasia and resulting in a higher concentration of various cells, including somatic stem cells, cytokines, and interleukins in the biosheet. To prepare biosheets containing more somatic stem cells, it is important to increase the aperture area on the template surface as much as possible.

The biosheet created with Y-shaped pore types was noticeably lighter in weight in the 3-week group than the other biosheets in any of the weeks and had insufficient fibrin and collagen tissue formation. The Y-shaped pore types in the 3-week group were embedded ventrally in the same individuals. Molds from a different week group were implanted on the opposite sides of those individuals; however, no similar changes were observed. Moreover, there were no problems with the biosheet created from another mold implanted on the ventral side of another individual. The reason for the inadequate formation of the biosheet created from the Y-shaped mold in the 3-week group is unclear, but it is possible that this was not due to a change in the implantation position or a specific individual but rather a sudden phenomenon. However, it may be necessary to further examine the relationship between implantation position and weight in individuals in the same week group.

In this study, biosheets were used at 2 weeks post-implantation because they were strong enough to cover the wound site, did not need to undergo collagen replacement, and contained abundant fibrin and somatic stem cells. For a biosheet to be used effectively at wound sites, it is important that it contains high levels of somatic stem cells and moderate collagen tissue as a scaffold, with a balance between the two. Further studies are required to determine the most effective and optimal implantation period for wound healing.

In the first week after implantation, no wound expansion was observed in the biosheet implantation group, whereas significant wound expansion was observed in the control group. Furthermore, the wound edges in the biosheet group adhered to the subcutaneous tissue from the first week after implantation but not until the second week in the control group. One reason for this may be that the biosheet, which has a certain degree of strength, was fixed around the entire periphery of the wound, which may have inhibited wound expansion. In addition, cytokines secreted by the somatic stem cells contained in the biosheet may have been involved. There have been many reports on the efficacy of MSCs in skin wound healing. During the inflammatory phase, exogenous bone marrow-derived mesenchymal stem cells (BMSCs) have been reported to preferentially migrate to sites of skin injury [23], and MSCs exert immunosuppressive effects at wound sites [4,24]. Furthermore, MSCs have also been reported to promote wound healing by changing the phenotypes of macrophages to the anti-inflammatory M2 phenotype, thereby promoting fibroblast proliferation and suppressing inflammation by inhibiting T cell proliferation [25]. In the proliferative phase, paracrine signaling from MSCs has been reported to promote wound healing, and MSCs have a potent secretome capable of influencing the activation, migration, and proliferation of various cells involved in the wound healing process as well as promoting angiogenesis, epithelialization, and fibroproliferation [4,24]. Somatic stem cells play an important role in wound healing, and the somatic stem cells in the biosheets could have influenced wound healing in this study. To clarify the effect of the biosheets on wound healing, it is necessary to further examine the gene expression of various cytokines and the differentiation of somatic stem cells into phenotypes typical of resident cutaneous cells at the site of wound healing in the biosheet and control groups.

The involvement of somatic stem cells in the wound healing process is important for chronic wounds that are difficult to heal because of many pathological conditions, such as burns, diabetes, and vascular insufficiency. Delayed wound healing has been implicated in infection, tissue hypoxia, necrosis, exudates, and excessive levels of pro-inflammatory cytokines [26]. Therefore, MSCs have been reported to contribute to wound healing in chronic wounds by directly attenuating the inflammatory response, and when MSCs are added to an active immune response, the inflammatory cytokines Tumor Necrosis Factor-α (TNF-α) and interferon-γ are reduced, and production of the anti-inflammatory cytokines IL-10 and IL-4 has been shown to increase [27]. Hypoxia activates MSCs [4]. In this study, there was a trend toward faster wound healing in the biosheet group than in the control group, but there were no statistically significant differences other than in the rate of wound contraction in the first week after implantation. These results were obtained in healthy dogs. Somatic stem cells in biosheets may contribute to wound healing in refractory chronic wounds caused by ischemia and diabetes.

This pilot study confirmed that the biosheet contains somatic stem cells that can be safely used in vivo and may be effective for wound healing. However, this study did not clarify the effects of the shape and size of the aperture in the surface of the mold on biosheet formation, particularly the balance between collagen formation and somatic stem cells, or the effects and mechanisms of somatic stem cells in enhancing the wound healing properties of biosheets. Further studies are required to clarify these issues in the future, as well as the appropriate duration for embedding, the shape of the mold and the nature of the holes in the mold surface, and the use of the mold in pathological disease models such as those of diabetes, and to provide a mechanistic analysis of the molecular process underlying the functions of biosheets in wound healing.

## Figures and Tables

**Figure 1 bioengineering-11-00435-f001:**
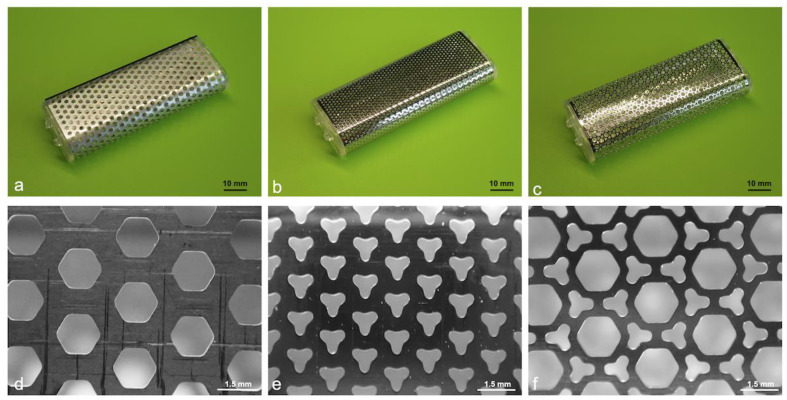
Overview and magnified images of the molds used for biosheet preparation. (**a**) O-shaped, (**b**) Y-shaped, and (**c**) O/Y mixed-pore molds. Surfaces of (**d**) O-shaped, (**e**) Y-shaped, (**f**) and O/Y mixed-pore molds.

**Figure 2 bioengineering-11-00435-f002:**
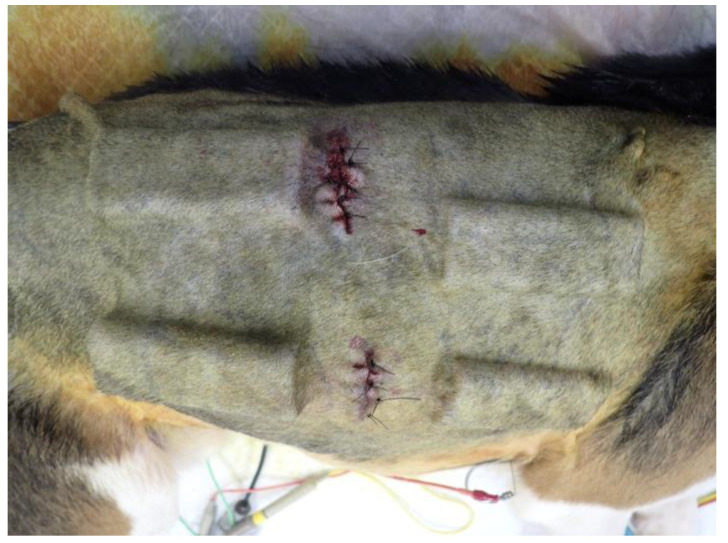
Photograph of embedded biosheet molds.

**Figure 3 bioengineering-11-00435-f003:**
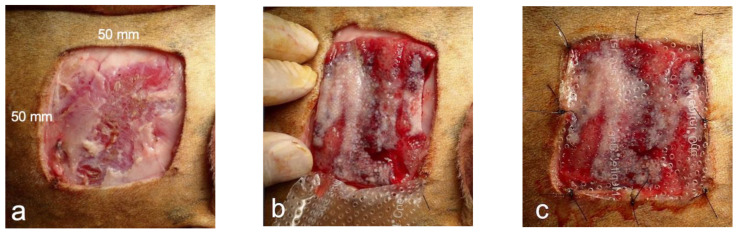
Images of biosheet attachment. (**a**) Preparation of skin defect (50 × 50 mm). (**b**) Covering of skin defect with biosheet. (**c**) Covering of the biosheet with a wound dressing material, which was fixed to the skin along with the dressing material.

**Figure 4 bioengineering-11-00435-f004:**
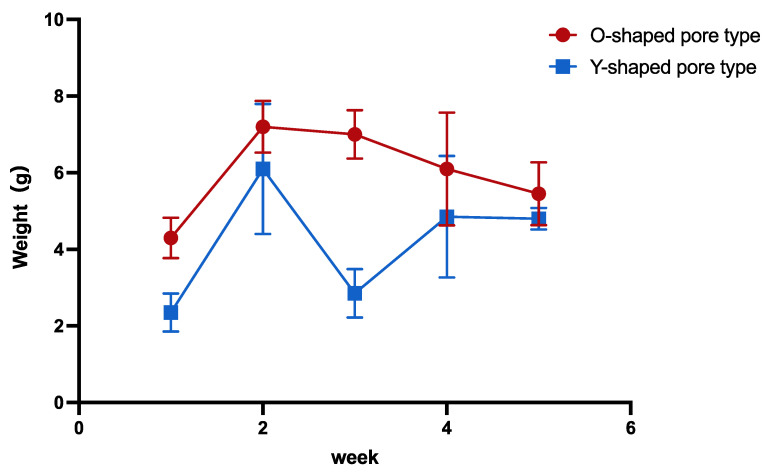
Comparison of the biosheet weights.

**Figure 5 bioengineering-11-00435-f005:**
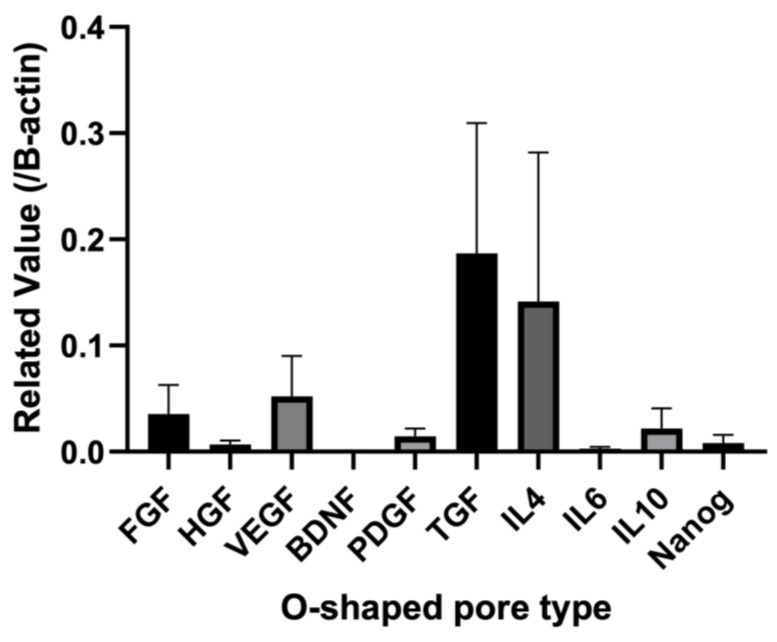
Quantitative RT-PCR expression levels of mRNAs for the biosheet prepared with the O-shaped pore mold after two weeks of implantation. Each value was normalized to beta-actin expression. FGF: fibroblast growth factor, HGF: hepatocyte growth factor, VEGF: vascular endothelial growth factor; BDNF: brain-derived neurotrophic factor; PDFG: platelet-derived growth factor. Mean ± SE, *n* = 4.

**Figure 6 bioengineering-11-00435-f006:**
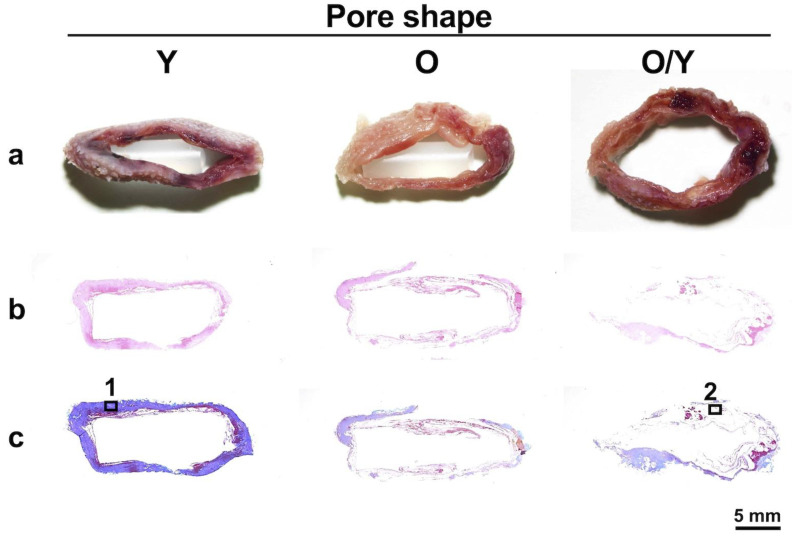
Macroscopic (**a**), H&E-stained (**b**), and Masson’s trichrome-stained (**c**) sections of biosheets formed in molds with different pore shapes. Collagen-rich tissue (**1**) and fibrin clot (**2**) positions formed in the molds with Y- and O/Y-shaped pores and their high-magnification images (**1-1**,**1-2**) and SSEA-4 and CD105 antigen-immunostained images (**2-1**,**2-2**).

**Figure 7 bioengineering-11-00435-f007:**
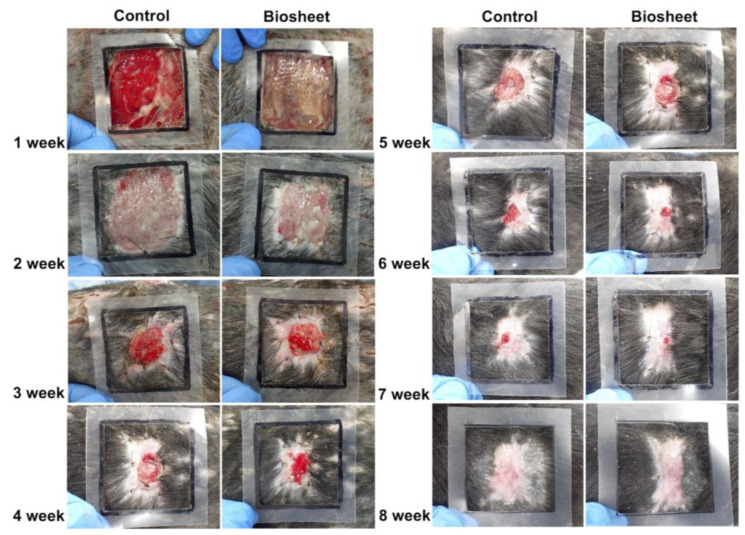
Macroscopic view of the wound healing process in the control and biosheet groups. Black squares represent the first skin defects (50 mm × 50 mm).

**Figure 8 bioengineering-11-00435-f008:**
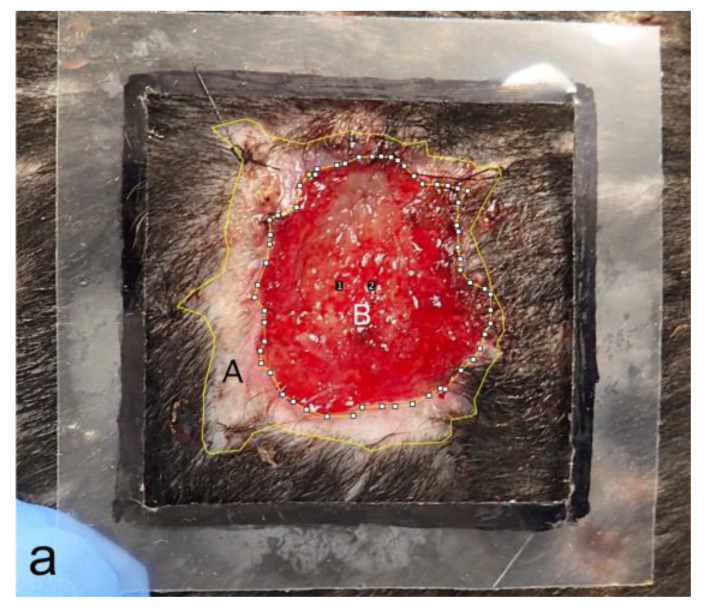
Comparison of the rates of wound contraction and epithelialization ratios. (**a**) Measured images. A: area within the wound margin, B: granulation tissue. Number 1 and 2 are automatically assigned when processing Image J. (**b**) Comparison of the rates of wound contraction. Rate of wound contraction at week 0 (25 cm^2^) −A/wound area at week 0 × 100. (**c**) Comparison of epithelialization ratios. Epithelialization ratio: epithelialized area (A and B)/A × 100. * *p* < 0.05.

**Table 1 bioengineering-11-00435-t001:** Primers used for qPCR.

Name	5′–3′	Direction	Position	GenBank No.
Bdnf-1	AATCCCATGGGTTACACGAA	sense	581	NM_001002975.1
Bdnf-2	GCCAGCCAATTCTCTTTTTG	anti-sense	707	NM_001002975.1
Fgf-1	CACTTCAAGGACCCCAAGAG	sense	190	XM_003432481.1
Fgf-2	ACAACGCCTCTCTCTTCTGC	anti-sense	332	XM_003432481.1
Hgf-1	ATGGGGAATGAGAAATGCAG	sense	1915	AB090353.1
Hgf-2	AAAAATGCCAGGACGATTTG	anti-sense	2124	AB090353.1
Pdgf-1	TCTTGGCAAGGCTTTTGTTT	sense	849	XM_539783.3
Pdgf-2	TTCCCTTATGGACACCGAGA	anti-sense	966	XM_539783.3
Vegf-1	CTACCTCCACCATGCCAAGT	sense	325	NM_001003175.2
Vegf-2	AGATGTCCACCAGGGTCTCA	anti-sense	458	NM_001003175.2
IL-4-1	CTCACCTCCCAACTGATTCC	sense	70	NM_001003159.1
IL-4-2	AGTCGTTTCTCGCTGTGAGG	anti-sense	202	NM_001003159.1
IL-6-1	GGCTACTGCTTTCCCTACCC	sense	108	U12234.1
IL-6-2	TTTTCTGCCAGTGCCTCTTT	anti-sense	305	U12234.1
IL-10-1	AGAACCACGACCCAGACATC	sense	300	U33843.1
IL-10-2	CCGCCTTGCTCTTATTCTCA	anti-sense	425	U33843.1
Nano-1	CCCAACTCTAGGGACCCTTC	sense	22	XM_543828
Nano-2	CAGATCCATGGAGGAAGGAA	anti-sense	156	XM_543828

**Table 2 bioengineering-11-00435-t002:** Area measurements. Values for each area are the averages of five measurements. A: area within the wound margin, B: granulation tissue.

	No. 1				No. 2			
	A (cm^2^)		B (cm^2^)		A (cm^2^)		B (cm^2^)	
	Biosheet	Control	Biosheet	Control	Biosheet	Control	Biosheet	Control
Week 0	25	25	0	0	25	25	0	0
Week 1	25.47	29.47	25.47	29.47	26.5	27.43	26.50	27.43
Week 2	17.03	23.27	14.3	20.55	13.18	19.93	10.35	17.04
Week 3	11.84	13.25	6.62	9.1	8.91	8.62	3.58	4.17
Week 4	8.61	8.32	3.58	4.15	5.45	5.2	1.31	1.97
Week 5	7.64	8.32	1.76	2.26	6.57	6.52	2.37	2.14
Week 6	5.81	5.23	0.6	0.36	5.15	5.91	0.36	0.83
Week 7	6.76	6.54	0	0	6.79	6.15	0.08	0.37
Week 8	7.23	7.41	0	0	6.25	6.78	0	0
	No. 3				No. 4			
	A (cm^2^)		B (cm^2^)		A (cm^2^)		B (cm^2^)	
	Biosheet	Control	Biosheet	Control	Biosheet	Control	Biosheet	Control
Week 0	25	0	0	0	25	25	0	0
Week 1	23.99	30.75	23.99	30.75	24.76	29.62	24.76	29.62
Week 2	17.59	17.59	15.72	14.91	14.97	19.86	10.48	15.25
Week 3	13.93	18.37	8.52	12.88	12.75	14.71	6.69	9.73
Week 4	7.92	14.26	3.55	7.63	11.30	11.63	4.11	5.10
Week 5	6.98	8.85	2.39	5.41	7.55	5.21	2.9	2.54
Week 6	5.12	5.69	0.84	2.65	4.19	3.7	1.1	1.26
Week 7	5.98	6.44	0.13	2.07	5.70	4.12	0.34	0.5
Week 8	6.18	5.65	0	1.36	6.31	4.74	0	0.16

## Data Availability

The raw data supporting the conclusion of this article will be made available by the authors without undue reservation. Request for access to these data should be made to Noritaka Maeta (n-maeta@ous.ac.jp).

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
