# Peer review of "Evaluation of Skin Wound Healing with Biosheets Containing Somatic Stem Cells in a Dog Model: A Pilot Study"

_bioengineering, 2024, doi:10.3390/bioengineering11050435_

Round 1
Reviewer 1 Report
Comments and Suggestions for Authors
This study presents a novel approach to enhancing mesenchymal stem cell (MSC) therapies for wound healing through the use of biosheets developed via in-body tissue architecture (iBTA). The incorporation of somatic stem cells into these biosheets aims to address the limitations of current MSC therapies, particularly in terms of cell engraftment and retention at the wound site. After my further consideration, there are still some problems with this manuscript. So I recommend this manuscript for publication after a major revision.
Comments:
1、The authors should fully reflect the dimensions of the biosheets and molds in the figures. I suggest adding a scale bar to Figure 1.
2、The authors do not seem to fully explain the reason for the significant decline in the quality of the Y-shaped biosheets between weeks 2 and 4 in Figure 4.
3、In Figure 5, there is excessive error in the TGF and IL4 measurements. Please provide a rationale for this large error while ensuring data accuracy.
4、Have the authors performed basic characterization of the physical and chemical properties of the obtained biosheets? It is challenging to observe specific information about the biosheets through macroscopic photographs and biological experiments alone.
5、In Figure 8a, it is recommended to provide area statistics. In 8b, the y-axis lacks units, which seem to be in percentage (%).
6、The authors have cited relatively few references. I suggest increasing the number of references, for example:
https://doi.org/10.1039/D3CS00923H
The degradable tissue repair elastomers have shown great potential in tissue regeneration.
The biosheets studied by the author, as materials with similar potential for tissue repair, could be highlighted in the introduction for their advantages over elastomers.
https://doi.org/10.1007/s10029-018-1799-8
The author appears to have conducted research on biosheets. Does this study present innovation and breakthroughs compared to that previous work?
7、The study suggests that the larger aperture area on the mold surface may allow more fluid to flow into the mold, facilitating the formation of fibrin and neoplasia, and resulting in a higher concentration of various cells, including somatic stem cells, cytokines, and interleukins in the biosheet. This is an interesting observation that ties the physical characteristics of the mold to the biological outcomes in the biosheet. Could you elaborate on the mechanistic relationship between the mold's aperture size and the cellular and molecular composition of the resulting biosheet? Additionally, how does this relationship influence the biosheet's efficacy in wound healing, particularly in terms of the balance between somatic stem cells and collagen tissue?
Comments on the Quality of English LanguageMinor editing of English language required.
Author Response
Manuscript ID: bioengineering-2958980
Title: Evaluation of Skin Wound Healing with Biosheets Containing Somatic Stem Cells in a Dog Model: A Pilot Study
Reviewer 1
Dear Reviewer,
Thank you for reviewing and providing constructive feedback for my manuscript. have addressed all of them and explained the changes below.”
1.The authors should fully reflect the dimensions of the biosheets and molds in the figures. I suggest adding a scale bar to Figure 1.
Response: Thank you for this valuable suggestion. Accordingly, we have added a scale bar to Figure 1.
2.The authors do not seem to fully explain the reason for the significant decline in the quality of the Y-shaped biosheets between weeks 2 and 4 in Figure 4.
Response: We thank you for raising this query and acknowledge that the reason could be better explained. Accordingly, we have added the following sentences:
Section 3. Results
Lines 214–215: A clear decrease in weight was noted in the Y-shaped pore types in the 3-week group.
Lines 226–227: The biosheet created with Y-shaped pore types showed poor formation of both fibrin and collagen tissues in the 3-week group.
Section 4. Discussion
Lines 362–373:The biosheet created with Y-shaped pore types was noticeably lighter in weight in the 3-week group than the other biosheets in any of the weeks and had insufficient fibrin and collagen tissue formation. The Y-shaped pore types in the 3-week group were embedded ventrally in the same individuals. Molds from a different week group were implanted on the opposite side of those individuals; however, no similar changes were observed. Moreover, there were no problems with the biosheet created from another mold implanted on the ventral side of another individual. The reason for the inadequate biosheet formation in the 3-week group created from the Y-shaped mold is unclear, but it is possible that this was not due to a change in the implantation position or a specific individual, but rather a sudden phenomenon. However, it may be necessary to further examine the relationship between implantation position and weight in individuals in the same weekly group.
3.In Figure 5, there is excessive error in the TGF and IL4 measurements. Please provide a rationale for this large error while ensuring data accuracy.
Response: Thank you for pointing out this issue. We believe that your point is valid. From our understanding, TGF could have increased to induce fibroblasts and IL4 could have increased as part of the immune response to the molds. However, the actual reasons underlying the increase could not be determined because of the small number of samples and the lack of temporal observations. Hence, I did not mention these in the paper. We plan to conduct more detailed analyses and investigate the molecular biological mechanism in the future.
4.Have the authors performed basic characterization of the physical and chemical properties of the obtained biosheets? It is challenging to observe specific information about the biosheets through macroscopic photographs and biological experiments alone.
Response: Thank you for your suggestion. We agree that basic characterization of biosheets is necessary before using them for extended applications. However, we could not perform them in the current study. In the future, we would like to conduct the basic characterization as well as a detailed molecular biological analysis of the biosheet.
5.In Figure 8a, it is recommended to provide area statistics. In 8b, the y-axis lacks units, which seem to be in percentage (%).
Response: According to your suggestion, we have included the area statistics for Figure 8a in Table 2. The units (%) to the y-axes of Figures 8b and 8c are also added.
- The authors have cited relatively few references. I suggest increasing the number of references, for example:
https://doi.org/10.1039/D3CS00923H
The degradable tissue repair elastomers have shown great potential in tissue regeneration.
The biosheets studied by the author, as materials with similar potential for tissue repair, could be highlighted in the introduction for their advantages over elastomers.
Response: According to your suggestion, we have added the following sentence:
- Introduction
Lines 71–74: Degradable biomedical elastomers (DBE) are useful materials for tissue repair that promote skin wound healing [13]. Although DBE is a useful scaffold for tissue repair, it does not contain somatic stem cells. Furthermore, the complexity of skin injuries necessitates that DBE be combined with other functional materials for multifunctional repair.
https://doi.org/10.1007/s10029-018-1799-8
The author appears to have conducted research on biosheets. Does this study present innovation and breakthroughs compare to that previous work?
Response: Fully collagenized biosheets without somatic stem cells for tissue repair were used in previous studies.
The current study uses biosheets before being fully collagenized. Biosheets before complete collagenization contain somatic stem cells, which are expected to not just act as a scaffold for tissue repair but also to play a facilitative role in wound healing.
We believe that the confirmation of the high content of somatic stem cells in the biosheet before collagenization and the verification of the wound healing effect using biosheet are innovative findings that can be considered as potential breakthroughs, compared to the findings from previous studies.
According to your suggestion, we have added the following sentence:
- Introduction
Lines 93–96: Fully collagenized biosheets were found effective for the repair of abdominal and bladder wall defects [19,20]; however, the effects of immature biosheets comprising somatic stem cells on tissue healing was not assessed.
7.The study suggests that the larger aperture area on the mold surface may allow more fluid to flow into the mold, facilitating the formation of fibrin and neoplasia, and resulting in a higher concentration of various cells, including somatic stem cells, cytokines, and interleukins in the biosheet. This is an interesting observation that ties the physical characteristics of the mold to the biological outcomes in the biosheet. Could you elaborate on the mechanistic relationship between the mold's aperture size and the cellular and molecular composition of the resulting biosheet? Additionally, how does this relationship influence the biosheet's efficacy in wound healing, particularly in terms of the balance between somatic stem cells and collagen tissue?
Response: Thank you for highlighting this feature. However, we have not performed in-depth research regarding the mechanistic details at this point and believe that it is a matter to be analyzed in the future.
Nevertheless, our thoughts regarding this point are as follows:
If the opening area of the pores on the surface of the mold is small, the speed of liquid penetration into the mold is low, and the amount of fibrin and neoplasm contained inside the mold is reduced compared to those inside a mold with a larger opening area for the same timing.
Because the fibrin and neoplasm inside the mold are considered to be replaced by collagen, we believe that less fibrin and neoplasm inside the mold can result in faster collagenization.
Furthermore, because somatic stem cells tend to decrease as collagenization proceeds molds with smaller apertures may tend to have fewer somatic stem cells than molds with larger apertures for the same timing.
The use of biosheets for skin wound healing requires that they contain a moderate amount of collagen for strength with many somatic stem cells. Hence, further studies on the mechanistic influences of the mold are needed to prepare biosheets that balance the amount of collagen with the number of somatic stem cells.

Reviewer 2 Report
Comments and Suggestions for Authors
Dear authors,
I read with great interest your manuscript titled “Evaluation of Biosheets Prepared from Different Mold Shapes for Wound Healing Applications”. The manuscript presents an investigation into the preparation and application of biosheets for wound healing using different mold shapes. The study explores the characteristics and efficacy of biosheets created with rectangular molds of various pore shapes, focusing on their composition, mechanical properties, and effectiveness in wound healing.
The methods section provides a comprehensive overview of the experimental procedures, including mold preparation, biosheet fabrication, and in vivo evaluation. However, specific details regarding the selection criteria for the study animals and the surgical procedures for biosheet implantation are lacking. Providing this information would enhance the reproducibility of the study and allow readers to better understand the experimental design. Please refer to ARRIVE guidelines, https://arriveguidelines.org/arrive-guidelines, for further information.
The findings of the research, including observations on the composition of the biosheet, its mechanical qualities, and the outcomes of wound healing, are succinctly summarised in the results section. More context and a discussion of possible mechanisms behind the observed impacts, however, could improve the interpretation of the data. For instance, the authors note that somatic stem cells are included in the biosheets, but they don't go into detail on how important these cells are for wound healing.
Important information on the study's significance for applications related to wound healing can be found in the discussion section. The authors may, however, go into more detail about the study's limitations and suggest future lines of inquiry to overcome them.
Overall, the manuscript presents an important contribution to the field of tissue engineering and wound healing. By evaluating biosheets prepared from different mold shapes, the study provides valuable insights into the factors influencing biosheet composition, mechanical properties, and efficacy in wound healing. Addressing the major and minor comments outlined above would further strengthen the manuscript and enhance its impact in the field.
With revisions addressing the comments provided, the manuscript has the potential to be suitable for publication in Bioengineering.
Author Response
Manuscript ID: bioengineering-2958980
Title: Evaluation of Skin Wound Healing with Biosheets Containing Somatic Stem Cells in a Dog Model: A Pilot Study
Reviewer 2
Dear Reviewer,
Thank you for reviewing and providing constructive feedback for my manuscript. have addressed all of them and explained the changes below.”
The methods section provides a comprehensive overview of the experimental procedures, including mold preparation, biosheet fabrication, and in vivo evaluation. However, specific details regarding the selection criteria for the study animals and the surgical procedures for biosheet implantation are lacking. Providing this information would enhance the reproducibility of the study and allow readers to better understand the experimental design. Please refer to ARRIVE guidelines, https://arriveguidelines.org/arrive-guidelines, for further information.
Response: Thank you for highlighting these shortcomings. Accordingly, we have added the following sentences:
Section 2. Materials and Methods
Lines 104–106: Dogs were used in this study because they have sufficient subcutaneous space for implantation of the molds and sufficient skin space to perform skin wound healing comparisons.
Line 120: aseptically
Lines 135–137: The skin was incised and the subcutaneous tissue was debrided using an electrocautery scalpel and forceps to create enough space for the molds.
Lines 138–139: The molds were embedded weekly for 5 weeks, and after 6 weeks, the skin was incised using an electrocautery scalpel, and the subcutaneous tissue was peeled away to remove all molds.
Line 183: using an electrocautery scalpel,
The findings of the research, including observations on the composition of the biosheet, its mechanical qualities, and the outcomes of wound healing, are succinctly summarized in the results section. More context and a discussion of possible mechanisms behind the observed impacts, however, could improve the interpretation of the data. For instance, the authors note that somatic stem cells are included in the biosheets, but they don't go into detail on how important these cells are for wound healing.
Response: Thank you for your suggestions. We have now revised the Discussion section to include more details.
Section 4. Discussion
Lines 398–400: Somatic stem cells play an important role in wound healing, and the somatic stem cells in the biosheets could have influenced wound healing in this study.
Important information on the study's significance for applications related to wound healing can be found in the discussion section. The authors may, however, go into more detail about the study's limitations and suggest future lines of inquiry to overcome them.
Response: According to your suggestion, we have added the following sentences:
Section 4. Discussion
Lines 419–423: However, this study did not clarify the effects of the shape and size of the aperture in the surface of the mold on biosheet formation, particularly the balance between collagen formation and somatic stem cells, the effects and mechanisms of somatic stem cells in enhancing the wound healing properties of biosheets.
Lines 423–427: Further studies are required to clarify these in the future, including the appropriate duration for embedding, shape of the mold and nature of holes on the mold surface, their use in pathological disease models such as those of diabetes, and a mechanistic analysis of the molecular process underlying the functions of biosheet in wound healing.

Round 2
Reviewer 1 Report
Comments and Suggestions for Authors
I have no more questions.
Comments on the Quality of English LanguageMinor editing of English language required.